# Characterization and Classification of Soils in Agroecosystems in a Moist Enclave in Northeastern Brazil

**Phâmella Kalliny Pereira Farias** [1], **Jeane Cruz Portela** [1,*], **Rafael Oliveira Batista** [1], **Joseane Dunga da Costa** [2], **Joaquim Emanuel Fernandes Gondim** [1], **Geisiane Xavier de Matos** [1], **Paulo Jardel Mota** [1], **Eulene Francisco da Silva** [1], **Francisco de Assis de Oliveira** [1], **Joaquim Odilon Pereira** [1], **Diego José da Costa Bandeira** [1], **Claudeone Manoel do Nascimento** [1], **Rauny Oliveira de Souza** [3], **Matias de Souza Dantas** [1], **Tiago da Costa Dantas Moniz** [1] and **Antonio Genilson Rodrigues Araujo** [1]

1   Department of Agronomic and Forest Science, Federal Rural University of the Semi-Arid—UFERSA, Mossoró 59625-900, Brazil
2   Department of Engineering and Technology, Federal Rural University of the Semi-Arid—UFERSA, Pau dos Ferros 59900-000, Brazil
3   Department of Botany and Zoology, Federal University of the Rio Grande do Norte—UFRN, Natal 59078-900, Brazil
*   Correspondence: jeaneportela@ufersa.edu.br

**Abstract:** Modern times have required studies that take into account the main soil features, aligning the use of land with the protection of more sensitive environments. From this perspective, this study aimed to perform a morphological description and determine the physical and chemical attributes for soil classification in the community of Poção, located in the municipality of Martins/RN, by highlighting the more sensitive attributes in the differentiation of environments through multivariate analysis. Nine soil profiles were identified to perform the morphological description and collect samples for physical and chemical analyses. The study updates the soil classes found in the study area: Acrisols, Planosols, and Cambisols, with the relief being the main factor responsible for the difference between soil attributes. The influence of organic matter on the soil attributes highlights the importance of its maintenance. Aluminum and the clay fraction are responsible for the distinction of the Acrisol class, whereas silt, potassium sodium, total organic carbon, the electrical conductivity of the saturation extract, and the cation exchange capacity allow the differentiation of Fluvisols.

**Keywords:** semi-arid; sustainable management; soil formation factors; pedogenesis; multivariate statistical analysis; land use; chemical attributes; physical attributes; morphological attributes

## 1. Introduction

The last few years have seen an increase in the search for information aimed at promoting the adequate use of land, especially in agroecosystems [1]. As a result, this scenario requires studies that consider the agricultural and environmental potential of the soil in order to align the use of land with the protection of more sensitive environments [2–4]. The Northeast region of Brazil shows climatic conditions that range from wet to semi-arid climates. For example, the state of Rio Grande do Norte contains mountain formations with different edaphoclimatic conditions that influence the formation of deeper and more acidic soils, justifying the importance of studying these regions in order to understand the local pedogenesis [5]. Moreover, there are areas with different rainfall and temperature patterns within zones considered semi-arid. These exceptional areas are called Caatinga moist-forest enclaves or 'brejos de altitude' [6].

The 'brejos de altitude' areas show an interaction between the geomorphological factor, represented by the relief, and the climate, favoring orographic precipitation [6]. From this perspective, the Serra de Martins mountains in the municipality of Martins-RN acts as a physical barrier that favors the development of orographic rainfall, responsible for the

higher precipitation volume in relation to the surrounding areas located in the Depressão Sertaneja lowlands [7].

Scientific studies on soil formation factors, especially climate and relief, in moist regions of northeastern Brazil favor the understanding of soil formation around the world and not only in Brazil, since soil characterization studies can solve problems related to physical, chemical, and morphological processes [8]. However, in the state of Rio Grande do Norte, studies involving moist-forest enclaves are still scarce, justifying the importance of studies aimed at updating the information about soils in 'brejos de altitude' located in northeastern Brazil.

Such studies enable the proper management of agroecosystems, reducing the problems caused by inadequate agricultural management, which would lead to the loss of the soil's production capacity and nutrient leaching [9]. Moreover, it should be noted that the different areas of soil science are integrated and not limited to traditional agriculture [10].

Scientific studies on the effects of soil on the landscape and their classification are important aspects to identify the potential and restrictions of different environments and act in the integrated management of actions that include strategies for sustainable agricultural planning in the moist-forest enclaves of northeastern Brazil [11]. These procedures are essential to understand the complex interactions of the water dynamics in the landscape and the influence of weathering on each environment [12,13].

The interaction between soil formation factors and processes gives origin to different soil classes, i.e., the relief variations, for example, directly influence the intensity of weathering, thus forming soils with different properties [14]. From this perspective, understanding the characteristics of soils in northeastern Brazil is important to prevent their degradation and assist in the process of change or planning in land use [12].

The municipality of Martins/RN shows a wide variability of soil classes, with the predominance of Ferralsols on the tops of elevations and Fluvisols and Leptosols in highland slopes and lowlands [15]. However, the soil classes need to be updated using a more detailed study in order to provide the adequate management of soils and agricultural areas.

Multivariate statistical techniques can be used to differentiate environments formed by different soil classes by gathering a lower number of variables and clustering samples according to their similarity or difference, showing effectiveness for soil studies [16].

The main hypotheses of the work include updating the soil classes in the study region and that the multivariate statistical technique better explains the existing distinctions due to the differences in soil attributes.

From this perspective, this study aimed to describe the morphology and determine the physical and chemical attributes for soil classification in the community of Poção, located in the municipality of Martins/RN, by highlighting the more sensitive attributes in the differentiation and characterization of environments using multivariate analysis.

## 2. Materials and Methods

### 2.1. Study Area

The study was conducted in the community of Poção, located in the municipality of Martins-RN, in the West Potiguar mesoregion and Umarizal microregion of the state of Rio Grande do Norte, at the following geographic coordinates: 6°05′16″ S and 37°54′40″ W. Although the area is located on the Borborema Plateau, its relief also comprises the Depressão Sertaneja lowlands, encompassing an area of 169.47 km$^2$ (Figure 1).

Even if though it is geographically contained within the Brazilian semi-arid region, the municipality of Martins shows different rainfall rates and mean temperatures in relation to its surrounding areas, which are inserted into the domains of the Depressão Sertaneja lowlands. Therefore, the Serra de Martins Mountain is considered a moist-forest enclave amidst the larger semi-arid domain, defined as a 'Brejo de Altitude' [7]. According to the Köppen classification, the local climate is classified as Aw, i.e., a rainy tropical climate with dry summers and a rainy season during the autumn [17].

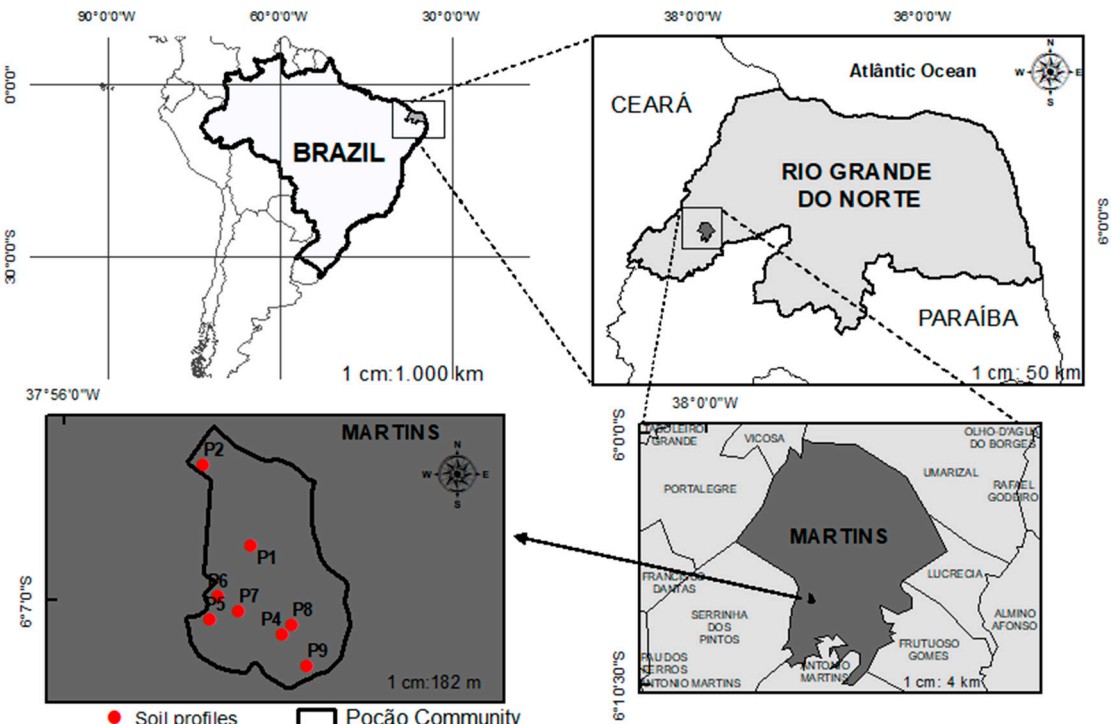

**Figure 1.** Location of the study area: Poção community in the municipality of Martins, Rio Grande do Norte, Brazil.

The climax vegetation of the Serra do Martins is a semi-deciduous forest [15]. The semi-deciduous nature of the forest over the plateau, contrasting with the hyperxerophilic vegetation of the Depressão Sertaneja lowlands, is due to the higher rainfall volume, the larger soil portion explored by plants, and the mild temperatures, which decrease transpiration.

Nine representative profiles of the study area were selected based on the exploratory survey of activities developed and their position in the landscape (Figure 2).

In area P1, minimal soil preparation was carried out with the addition of organic matter of plant and animal origin from the property. In the dry period, the following crops were sown: Cowpea (*Vigna unguiculata* L. Walp), Corn (*Zea mays* L.), and Jerimum/Squash (*Curcubita pepo* L.), and, in the rainy period, the area was destined for the production of vegetables at the highest elevation of the land. The commercialization of agricultural production takes place in the city. In area P2, farmers use the soil to make earthenware pieces, depending on their appropriate characteristics, providing a source of income for families. P3 has a history of minimal soil preparation for planting crops such as beans, corn, and pumpkin, during the dry period, with the aid of irrigation.

P4, P7, and P9 are under preserved native vegetation (humid enclave Caatinga). P5 and P6 are in areas where there is minimal soil preparation, involving contour lines to control the erosion process with subsequent planting of vegetables that are sold in the city and widely used in local commerce. P8 is a fallow area, being prepared for future use with the planting of Corn (*Zea mays* L.), Fava Beans (*Vicia faba* L.), Cassava (*Manihot esculenta* L.), and Pumpkin (*Curcubita pepo* L.).

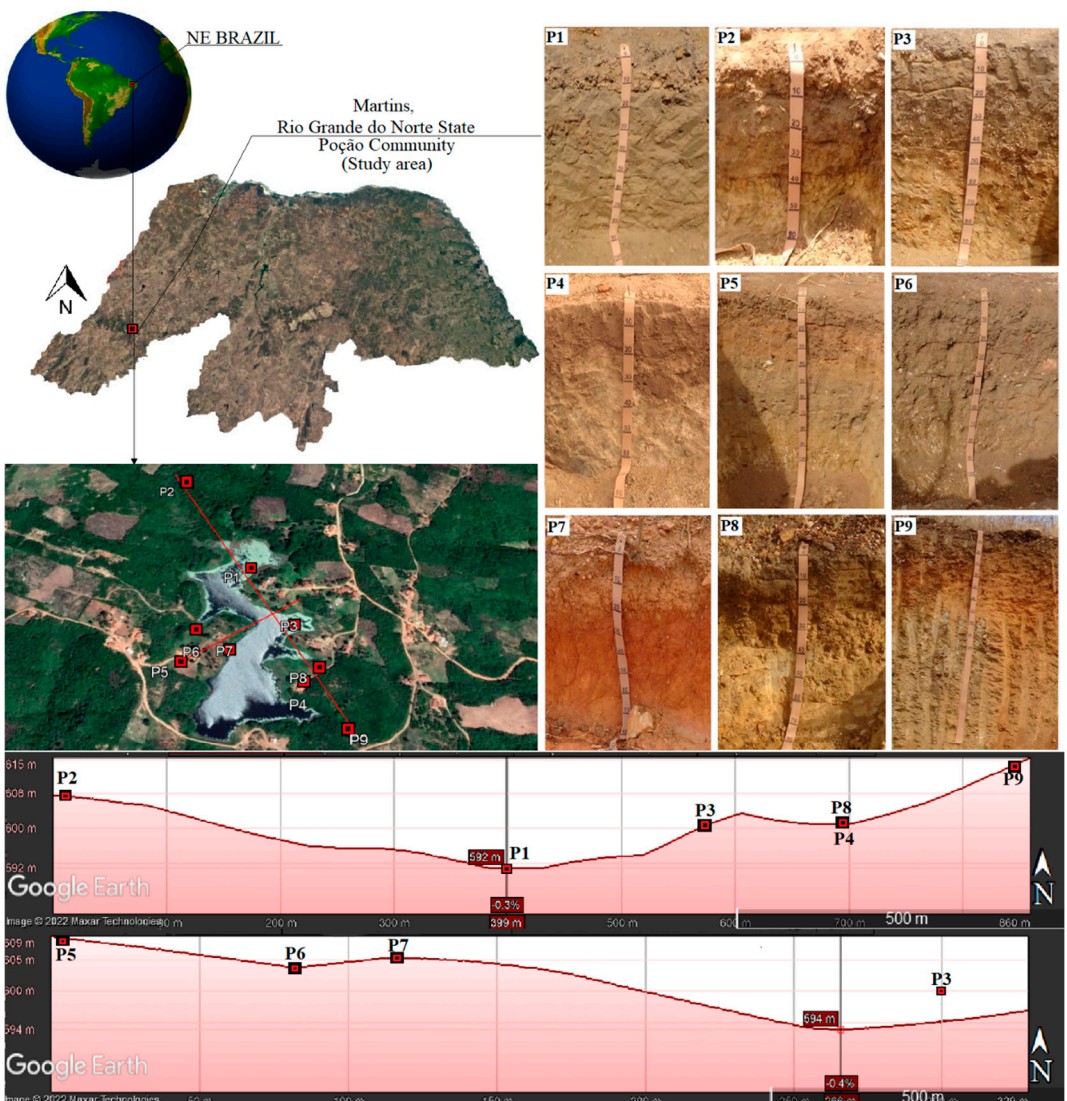

**Figure 2.** Representative profiles of the study area. **Note:** Altitude = P1—NEOSSOLO FLÚVICO Ta Eutrófico típico (Fluvisols)/594 m; P2—CAMBISSOLO HÁPLICO Ta Eutrófico típico (Cambisols)/606 m; P3—NEOSSOLO FLÚVICO Ta Eutrófico típico (Fluvisols)/598 m; P4—LUVISSOLO CRÔMICO Órtico típico (Luvisols)/603 m; P5—PLANOSSOLO HÁPLICO Eutrófico típico (Planosols)/608 m; P6—NEOSSOLO LITÓLICO Chernossólico típico (Leptsols)/603 m; P7—ARGISSOLO VERMELHO-AMARELO Eutrófico típico (Acrisols)/605 m; P8—PLANOSSOLO HÁPLICO Eutrófico típico (Planosols)/602 m; and P9—NEOSSOLO LITÓLICO Eutrófico típico (Leptsols)/613 m.

## 2.2. Sample Collection

The soil profiles were described and collected in all soil horizons according to reference [18] and classified up to the 4th level based on the Brazilian Soil Classification System [19] and its correspondence with the World Reference Base [20]. The samples were collected in triplicate to perform physical and chemical analyses.

Soil samples with deformed and undeformed structures were collected from all horizons in the representative soil profiles of the study area. The deformed samples were air-dried, ground, and passed through 20 mm and 2 mm mesh size sieves, thus obtaining the gravel (>2.00 mm to <20.00 mm) and air-dried fine portions (ADF) (<2.00 mm), according to reference [21]. The undeformed samples were collected in triplicate in all horizons using volumetric rings (5.0 cm in height and 5.0 cm in diameter), totaling 105 samples (3 samples × 35 horizons).

The physical and chemical analyses were performed using the air-dried fine samples (ADF) in three replications at the Laboratory of Soil Physics and Management of the Center of Agricultural Sciences (CCA) of the Federal Rural University of the Semi-arid Region—UFERSA.

### 2.3. Soil Analyzes

The physical analyses performed consisted of particle size analysis using the pipette method using a chemical dispersant (sodium hexametaphosphate) and distilled water in 20 g of TSFA, with slow mechanical agitation using a stirrer (Wagner 50 rpm) for 16 h [21]. The sand portion (2 to 0.05 mm) was quantified by sieving, the clay portion (<0.002 mm) was quantified by sedimentation, and the silt portion (0.5 to 0.002 mm) was quantified by the difference between the sand and clay portions. Based on these data, textural classification was carried out using the textural triangle; the soil density was determined using the volumetric ring method; and particle density was determined using a volumetric flask with alcohol by considering only the soil volume effectively occupied by the particles and not the porous spaces [21].

The chemical analyses consisted of: pH in water and KCl (1:2.5); electrical conductivity of the saturation extract (EC); extraction of available P and $Na^+$ and $K^+$ with $Mehlich^{-1}$, and determination of the available P contents using colorimetry and $Na^+$ and $K^+$ using flame photometry. The extraction of the exchangeable cations $Ca^{2+}$, $Mg^{2+}$, and $Al^{3+}$ was performed with potassium chloride, and the contents were determined by titration. H + Al were extracted with 1 mol $L^{-1}$ calcium acetate at pH 7.0 and determined by volumetric titration with a solution of 0.025 mol $L^{-1}$ NaOH [22]. The total organic carbon (TOC) was determined by titration of the potassium dichromate (0.167 mol $L^{-1}$) remaining after the wet oxidation process according to the methodology proposed by the authors of reference [22].

Based on the analyses performed, the following indices were obtained: sum of bases (SB); effective cation exchange capacity (t); cation exchange capacity at pH 7.0 (CEC); base saturation (V%); saturation by exchangeable aluminum (m%); and the percentage of exchangeable sodium (PES), determined according to the authors of reference [21].

### 2.4. Statistical Analysis

A statistical analysis of the soil profiles studied was performed using multivariate techniques and the software Statistica 7.0 [23] following Pearson's method ($p \leq 0.05$) for the 16 variables in order to ensure that these attributes had minimum correlations that could justify their use in the data matrix. The correlation matrix established a pattern for the analytical results to apply multivariate techniques such as Factor Analysis (FA) and Principal Component Analysis (PCA) [24].

In factor analysis, the principal components that showed eigenvalues higher than 1 were extracted, and the factorial axes were rotated using the Varimax method. The value of 0.70 was established to consider the significant factorial loads [24].

Two diagrams were constructed for the principal component analysis (factors 1 and 2) for the chemical and physical attributes. Based on these data, a two-dimensional diagram was constructed to depict the distinction of areas, and another for vector projection to verify the sensitive soil attributes in the differentiation of the study area [24]. The physical variables involved consisted of: TOC; pH ($H_2O$); EC; P; $K^+$; $Na^+$; $Ca^{2+}$; $Mg^{2+}$; $Al^{3+}$; (H + Al); SB; CEC; and V.

## 3. Results and Discussion

### 3.1. Soil Classification

In the profiles described, the relief formation factor influenced the differentiation of soil classes. According to reference [15], the main soil classes found in the study area are Ferralsols, Luvisols, and Leptsols. However, the present study also identified the Acrisol, Planosol, and Cambisol soil classes.

With regard to the taxonomic classification, profiles 1 and 3 were classified as NEOSSOLO FLÚVICO Ta Eutrófico típico for being little-developed, derived from alluvial sediments with particle size variations in depth, showing a high-activity clay fraction and base saturation ≥ 50% within 150 cm from the surface, and not fitting the other classes in order to be classified at the 4th level [19]. According to reference [20], profiles 1 and 3 were classified as Fluvisols.

Profile 2 was classified as a CAMBISSOLO HÁPLICO Ta Eutrófico típico, since it is formed by mineral material with an incipient B horizon subjacent to any surface horizon (except the histic horizon with 40 cm or more in depth), does not fit the other classes in order to be classified at the 2nd level, shows a high-activity clay fraction and base saturation ≥ 50%, both in most of the first 100 cm of the B horizon (including BA), and does not fit the other classes in order to be classified at the 4th level [19]. According to reference [20], profile 2 was classified as Cambisols.

Profile 4 was classified as a LUVISSOLO CRÔMICO Órtico típico, since it consists of mineral material, showing a textural B horizon with a high-activity clay fraction and high base saturation in most of the first 100 cm of the B horizon (including BA), immediately below any A horizon (except A chernozemic) or under an E horizon and with a chromic character in most of the first 100 cm of horizon B (including BA) [19]. For reference [20], profile 4 was classified as Luvisols.

Profiles 5 and 8 were classified as PLANOSSOLO HÁPLICO Eutrófico típico, since they are formed by mineral material with an A horizon followed by a flat B horizon, not fitting the other classes in order to be classified at the 2nd level, with base saturation ≥ 50% in most of the B horizon (including BA or BE) within 150 cm from the surface, and not fitting the other classes in order to be classified at the 4th level [19]. According to reference [20], profiles 5 and 8 follow the same classification, i.e., Planosols.

Profile 6 was classified as NEOSSOLO LITÓLICO Chernossólico típico for also being a little-developed soil with lithic contact within 50 cm from the surface, with the presence of a chernozemic horizon and clay activity ≥ 20 cmolc kg$^{-1}$ in most of the C horizon (including CA) within 50 cm from the soil surface without a carbonatic character [19]. According to reference [20], the classification correspondence for profile 6 is Leptsols.

Profile 7 was classified as ARGISSOLO VERMELHO-AMARELO Eutrófico típico, corresponding to a soil composed of mineral material, with a textural B horizon immediately below the A or E horizons, low-activity clay, and red-yellowish or yellow-reddish colors that do not fit the previous classes of the 2nd level, not fitting the other classes to be classified at the 4th level [19]. According to reference [20], profile 7 was classified as Acrisols.

Profile 9 was classified as a NEOSSOLO LITÓLICO Eutrófico típico since, similar to profile 6, it is also a little-developed soil with lithic contact, differing from profile 6 for showing base saturation ≥ 50% in most of the horizons within 50 cm of the surface, and not fitting the other classes in order to be classified at the 4th level [19]. For reference [20], profile 9 was classified as Leptsols.

### 3.2. Morphological Characterization

The different soil classes found in the study area result from the action of soil formation factors associated with pedogenetic processes, thus showing variations in the morphological features (Table 1).

The profiles corresponding to Fluvisols and Leptsols (P1, P3, P6, and P9) did not differ regarding the color of the horizons (Table 1), with a 2.5 Y hue and showing a yellow color. With regard to the value and chroma, there were lighter and grayer colors.

The Cambisol profile (P2) showed a more reddish, 5 YR hue without variations across the horizons. This profile shows a structure in subangular blocks with a moderate development degree and a small size, except for the diagnostic Bi horizon, which possesses a strong development degree and a structure with angular blocks resulting from a more intense pedogenetic process. The diagnostic horizon also differs from the others for showing

a very hard consistency when dry and a firm consistency when moist due to the higher clay content found in this subsurface horizon (Table 1).

**Table 1.** Morphological attributes of the soil profiles of the Poção community, Martins/RN.

| Hor./Depth (cm) | Alt. (m) | Color Munsell | | Parent Material | Structure | Consistency | | | Trans. |
|---|---|---|---|---|---|---|---|---|---|
| | | Dry | Moist | | | Dry | Moist | Wet | |
| *Profile 1—NEOSSOLO FLÚVICO Ta Eutrófico típico (Fluvisols)* | | | | | | | | | |
| Ap (0–15) | | 2.5 Y 6/4 | 2.5 Y 4/4 | | 2 P/M Bls | MD | MFi | NPl/NPe | ap |
| 2C1 (15–34) | | 2.5 Y 5/4 | 2.5 Y 4/4 | | 1 P/M Bls | D | F | LgPl/LgPe | go |
| 2C2(34–55) | 595 | 2.5 Y 5/4 | 2.5 Y 4/4 | Arenite Paleogene | 1 M/G Bls | D | F | LgPl/LgPe | do |
| 3C3 (55–65) | | 2.5 Y 5/4 | 2.5 Y 4/4 | | 3 M/G Bls | D | MFi | NPl/NPe | go |
| 3C4 (65–85) | | 2.5 Y 5/4 | 2.5 Y 4/4 | | 4 M Bls | D | MF | NPl/NPe | go |
| *Profile 2—CAMBISSOLO HÁPLICO Ta Eutrófico típico (Cambisols)* | | | | | | | | | |
| A (0–10) | | 5 YR 4/3 | 5 YR 3/3 | | 3 P Bls | D | F | NPl/NPe | co |
| AB (10–21) | 604 | 5 YR 4/3 | 5 YR 3/3 | Arenite | 3 P Bls | D | F | NPl/NPe | co |
| BA (21–40) | | 5 YR 4/2 | 5 YR 3/2 | Paleogene | 3/4 P Bls | D | F | NPl/LgPe | ao |
| B (40–60) | | 5 YR 4/2 | 5 YR 3/2 | | 4 P Bla | MD | Fi | NPl/LgPe | ao |
| *Profile 3—NEOSSOLO FLÚVICO Ta Eutrófico típico (Fluvisols)* | | | | | | | | | |
| A1 (0–10) | | 2.5 Y 5/3 | 2.5 Y 4/3 | | 3 MP Bls | D | F | NPl/LgPe | co |
| A2 (10–20) | 599 | 2.5 Y 5/3 | 2.5 Y 4/3 | Arenite | 3 MP Bls | D | F | NPl/LgPe | dp |
| AC (20–50) | | 2.5 Y 4/3 | 2.5 Y 3/3 | Paleogene | 3 MP Bls | D | F | LgPl/LgPe | go |
| C (50–98) | | | | | | | | | |
| *Profile 4—LUVISSOLO CRÔMICO Órtico típico (Luvisols)* | | | | | | | | | |
| A (0–10) | | 5 Y 5/3 | 5 Y 4/3 | | 3 P Bls | D | F | NPl/NPe | co |
| AB (10–20 | | 5 Y 4/3 | 5 Y 3/3 | | 3 P Bls | D | F | NPl/LgPe | go |
| BA (20–33) | 601 | 5 Y 4/3 | 5 Y 3/3 | Arenite Paleogene | 3 MP/P Bls | MD | F | NPl/LgPe | dp |
| Bt (33–57) | | 7.5 YR 5/4 | 7.5 YR 4/4 | | 3 MP/P Bls | MD | F | LgPl/Lg | dp |
| BC (57–65) | | 7.5 YR 5/4 | 7.5 YR 4/4 | | 3 P Bls | MD | F | NPl/LgPe | co |
| *Profile 5—PLANOSSOLO HÁPLICO Eutrófico típico (Planosols)* | | | | | | | | | |
| A1 (0–10) | | 10 YR 4/4 | 10 YR 3/4 | | 2 P Gr | LgD | MF | NPl/LgPe | cp |
| A2 (10–32) | | 10 YR 4/4 | 10 YR 3/4 | Arenite | 2 MP Gr | D | F | NPl/NPe | cp |
| A3 (32–44) | 607 | 10 YR 4/6 | 10 YR 3/6 | Paleogene | 2 MP/P Bls | D | F | NPl/LgPe | dp |
| BA (44–64) | | 10 YR 4/3 | 10 YR 3/3 | | 2/3 MP/P Bla/Bls | D | F | LgPl/LgPe | ao |
| Bt (64–100) | | 10 YR 4/3 | 10 YR 3/3 | | 4 P/M Bla/Bls | D | F | LgPl/LgPe | ao |
| *Profile 6—NEOSSOLO LITÓLICO Chernossólico típico (Leptsols)* | | | | | | | | | |
| A1 (0–28) | | 2.5 Y 4/2 | 2.5 Y 3/2 | Arenite | 3 MP Bls | MD | F | LgPl/LgPe | co |
| A2 (28–44) | 605 | 2.5 Y 4/2 | 2.5 Y 3/2 | Paleogene | 3 MP Bls | D | F | LgPl/LgPe | go |
| CR (44–90) | | | | | | | | | |
| *Profile 7—ARGISSOLO VERMELHO-AMARELO Eutrófico típico (Acrisols)* | | | | | | | | | |
| A (0–10) | | 10 YR 4/6 | 10 YR 3/6 | | 2 P Bls | MD | F | NPl/NPe | co |
| BA (10–20) | 606 | 10 YR 4/6 | 10 YR 3/6 | Arenite | 3 MP/P Bls | MD | F | NPl/NPe | co |
| Bt (20–50) | | 5 YR 5/6 | 5 YR 4/6 | Paleogene | 4 MP Bla | D | F | LgPl/LgPe | dp |
| BC (50–75) | | 5 YR 5/6 | 5 YR 4/6 | | 4 MP/P Bla | D | F | LgPl/LgPe | dp |
| *Profile 8—PLANOSSOLO HÁPLICO Eutrófico típico (Planosols)* | | | | | | | | | |
| A (0–6) | | 2.5 Y 4/3 | 2.5 Y 3/3 | | 2 P Bls/Gr | D | F | LgPl/LgPe | co |
| BA (6–23) | 602 | 2.5 Y 4/3 | 2.5 Y 3/3 | Arenite | 2 P Bls | D | F | LgPl/LgPe | ao |
| Bt (23–43) | | 2.5 Y 5/4 | 2.5 Y 4/4 | Paleogene | 4 P Bla | D | F | LgPl/Pe | go |
| CR (43–60) | | | | | | | | | |
| *Profile 9—NEOSSOLO LITÓLICO Eutrófico típico (Leptsols)* | | | | | | | | | |
| A (0–20) | 614 | 2.5 Y 5/3 | 2.5 Y 4/3 | Arenite Paleogene | 2 MP/P Bls | MD | F | NPl/NPe | co |

Structure: 1 = weak, 2 = moderate, P = small, M = medium, G = large, Gr = granular; Bla = angular blocks; Bls = subangular blocks; Consistency: Ma = soft, D = hard, Fr = friable, Fi = firm, N = No, M = very, Lg = slightly, Pl = plastic, Pe = Sticky; T = Transition, p = flat, d = diffuse, c = clear, g = gradual.

The Luvisol profile (P4) shows a clear differentiation between the A and Bt horizons due to the texture and color contrast between them (Table 1). The color along the profile

has a hue ranging from 5 Y to 7.5 YR. In all horizons, the structure showed a moderate development degree, with a class ranging from medium to small and a sub-angular block structure, corroborating the results found by the authors of reference [25] when performing the physical, chemical, mineralogical, and morphological characterization of Luvisols.

Across the profile, the Planosols showed no color variation (Table 1). However, it is possible to see a variation in the development degree, ranging from weak (in the surface horizons) to strong (in the subsurface horizons), which could be related to textural changes identified across the horizons. The authors of ref. [26], in their study on the pedogenesis of Planosols, also evidenced a clear and abrupt textural change highlighted by a texture contrast across the profile.

The Acrisol profile (P7) showed a depth variation in relation to all morphological attributes (Table 1). The surface horizons have a 10 YR hue, a structure in sub-angular blocks with weak to moderate development, a very hard consistency when dry, friable when moist, and non-plastic and non-sticky when wet. The subsurface horizons, in turn, have a 5 YR hue, a structure in angular blocks with a strong development degree, a hard consistency when dry, friable when moist, and slightly plastic and sticky when wet. All this variation stems from the textural change within the profile, which is characteristic of the formation process of this soil class, evidenced by the translocation of the clay fraction in depth.

### 3.3. Physical Characterization

In the particle size composition of the Fluvisols and Leptsols in general, the total sand fraction predominated in relation to the silt and clay fractions across the horizons, with contents ranging from 380 g/kg in the A1 horizon of P3 to 800 g/kg in the C horizon of P3 (Table 2).

The particle size distribution of the Cambisol (P2) showed little variation, and the total sand content decreased with the depth (Table 2), ranging from 640 g/kg in the AB horizon to 600 g/kg in the B horizon. On the other hand, the inverse occurred with the clay fraction as the contents increased with the depth, ranging from 220 g/kg in the A horizon to 230 g/kg in the B horizon. According to reference [27], the little variation in the clay content across the profile is a characteristic of this type of soil.

In the Luvisol profile (P4), the sand fraction predominated over silt and clay, with values ranging from 620 g/kg in the AB horizon to 490 g/kg in the Bt horizon (Table 2). The clay contents ranged from 130 g/kg in the A horizon to 310 g/kg in the Bt horizon, highlighting the bissialitization process along with the mobilization of clay from surface to subsurface horizons [19]. Soil texture was influenced by the higher contents of the clay fraction, which, according to reference [28], improves soil aggregation through the contact between particles.

The Planosols profiles (P5 and P8) showed textures ranging from sandy loam to sandy clay loam, with sand contents ranging from 710 g/kg to 630 g/kg (in the horizons of P5) and 730 g/kg to 610 g/kg (in the horizons of P8) (Table 2). The clay content increased with the depth, and these values were responsible for the abrupt textural change that occurs in this soil class.

In the Acrisol (P7), the sand contents were higher in the A and BA horizons (600 g/kg and 650 g/kg, respectively), with a subsequent increase with depth in horizons Bt and BC (200 g/kg and 250 g/kg, respectively) (Table 2). The clay content increases from 170 g/kg and 210 g/kg (in the BA and A horizons, respectively) to 570 g/kg and 610 g/kg (in the BC and Bt horizons, respectively).

Particle density in the soil profiles analyzed ranged from 2.25 to 2.61 g/cm$^3$ (Table 2). According to reference [29], the low particle density values observed in mineral soils can be explained by the presence of organic matter (15% to 20%) and the lower soil clay content in the soil, with low specific mass values, thus influencing the density values of particles smaller than 2.40 kg/dm$^3$. Overall, the soil density values were considered low, ranging from 1.2 g/cm$^3$ to 1.6 g/cm$^3$ (Table 2). Similar to particle density, these low values could be

related to the minimum management of the planting system adopted in these areas, which contributes to maintaining and adding organic matter.

**Table 2.** Physical attributes of the soil profiles of the Poção community, Martins/RN.

| Hor./Depth | Gravel | Sand Total | Silt | Clay | Silt/ Clay | Textural Class (SiBCS) | PD | SD |
|---|---|---|---|---|---|---|---|---|
| cm | % | --------------g/kg------------- | | | | | ----g/cm³---- | |
| Profile 1—NEOSSOLO FLÚVICO Ta Eutrófico típico (Fluvisols) | | | | | | | | |
| Ap (0–15) | 1.2 | 390 | 340 | 270 | 1.26 | Loam | 2.25 | 1.20 |
| 2C1 (15–34) | 4.3 | 600 | 210 | 190 | 1.10 | Sandy loam | 2.40 | 1.20 |
| 2C2(34–55) | 1.3 | 620 | 210 | 170 | 1.23 | Sandy loam | 2.39 | 1.20 |
| 3C3 (55–65) | 2.4 | 620 | 180 | 200 | 0.90 | Sandy clay loam | 2.49 | 1.60 |
| 3C4 (65–85) | 3.0 | 620 | 150 | 230 | 0.65 | Sandy clay loam | 2.50 | 1.60 |
| Profile 2—CAMBISSOLO HÁPLICO Ta Eutrófico típico (Cambisols) | | | | | | | | |
| A (0–10) | 2.9 | 620 | 160 | 220 | 0.73 | Sandy clay loam | 2.52 | 1.40 |
| AB (10–21) | 4.4 | 640 | 160 | 200 | 0.80 | Sandy clay loam | 2.45 | 1.10 |
| BA (21–40) | 6.9 | 620 | 160 | 220 | 0.73 | Sandy clay loam | 2.49 | 1.40 |
| B (40–60) | 5.8 | 600 | 170 | 230 | 0.74 | Sandy clay loam | 2.47 | 1.50 |
| Profile 3—NEOSSOLO FLÚVICO Ta Eutrófico típico (Fluvisols) | | | | | | | | |
| A1 (0–10) | 5.5 | 380 | 390 | 230 | 1.70 | Loam | 2.29 | 1.20 |
| A2 (10–20) | 8.5 | 630 | 210 | 160 | 1.31 | Sandy loam | 2.43 | 1.60 |
| AC (20–50) | 13.2 | 630 | 180 | 190 | 0.95 | Sandy loam | 2.54 | 1.20 |
| C (50–98) | 5.9 | 800 | 110 | 90 | 1.22 | Loamy sand | 2.52 | 1.20 |
| Profile 4—LUVISSOLO CRÔMICO Órtico típico (Luvisols) | | | | | | | | |
| A (0–10) | 33.0 | 590 | 280 | 130 | 2.15 | Sandy loam | 2.30 | 1.50 |
| AB (10–20) | 16.3 | 620 | 230 | 150 | 1.53 | Sandy loam | 2.34 | 1.60 |
| BA (20–33) | 7.9 | 490 | 210 | 290 | 0.72 | Sandy clay loam | 2.47 | 1.50 |
| Bt (33–57) | 10.5 | 490 | 200 | 310 | 0.64 | Sandy clay loam | 2.30 | 1.60 |
| BC (57–65) | 8.9 | 560 | 210 | 230 | 0.91 | Sandy clay loam | 2.49 | 1.40 |
| Profile 5—PLANOSSOLO HÁPLICO Eutrófico típico (Planosols) | | | | | | | | |
| A1 (0–10) | 11.6 | 660 | 220 | 120 | 1.83 | Sandy loam | 2.31 | 1.20 |
| A2 (10–32) | 10.0 | 660 | 180 | 160 | 1.12 | Sandy loam | 2.37 | 1.30 |
| A3 (32–44) | 3.8 | 630 | 190 | 180 | 1.05 | Sandy loam | 2.43 | 1.40 |
| BA (44–64) | 7.7 | 710 | 150 | 140 | 1.07 | Sandy loam | 2.63 | 1.70 |
| Bt (64–100) | 10.8 | 630 | 140 | 230 | 0.61 | Sandy clay loam | 2.56 | 1.40 |
| Profile 6—NEOSSOLO LITÓLICO Chernossólico típico (Leptsols) | | | | | | | | |
| A1 (0–28) | 10.4 | 630 | 240 | 130 | 1.85 | Sandy loam | 2.37 | 1.40 |
| A2 (28–44) | 10.8 | 640 | 210 | 150 | 1.40 | Sandy loam | 2.49 | 1.20 |
| CR (44–90) | 14 | 700 | 150 | 150 | 1.00 | Sandy loam | 2.52 | 1.60 |
| Profile 7—ARGISSOLO VERMELHO-AMARELO Eutrófico típico (Acrisols) | | | | | | | | |
| A (0–10) | 10.5 | 600 | 190 | 210 | 0.90 | Sandy clay loam | 2.51 | 1.20 |
| BA (10–20) | 15.9 | 650 | 180 | 170 | 1.06 | Sandy loam | 2.44 | 1.20 |
| Bt (20–50) | 2.7 | 200 | 190 | 610 | 0.31 | Clay | 2.61 | 1.20 |
| BC (50–75) | 2.4 | 250 | 180 | 570 | 0.62 | Clay loam | 2.58 | 1.20 |
| Profile 8—PLANOSSOLO HÁPLICO Eutrófico típico (Planosols) | | | | | | | | |
| A (0–6) | 30.1 | 670 | 230 | 100 | 2.30 | Sandy loam | 2.51 | 1.30 |
| BA (6–23) | 4.9 | 610 | 200 | 190 | 1.05 | Sandy loam | 2.49 | 1.30 |
| Bt (23–43) | 2.0 | 640 | 160 | 200 | 0.80 | Sandy clay loam | 2.59 | 1.30 |
| CR (43–60) | 4.5 | 730 | 130 | 140 | 0.93 | Sandy loam | 2.59 | 1.30 |
| Profile 9—NEOSSOLO LITÓLICO Eutrófico típico (Leptsols) | | | | | | | | |
| A (0–20) | 7.9 | 700 | 190 | 110 | 1.73 | Sandy loam | 2.53 | 1.30 |

PD = particle density; SD = soil density.

*3.4. Chemical Characterization*

All the soil profiles evaluated had an acidic reaction, with water pH values ranging from 3.98 to 6.70 (Table 3).

**Table 3.** Chemical attributes of soil profiles in the Poção community, Martins/RN.

| Hor./Depth | TOC | pH (1:2.5) | | ΔpH | EC | P | K+ | Na+ | Ca2+ | Mg2+ | Al3+ | (H + Al) | SB | t | CEC | Clay Activity | V | m | PST |
|---|---|---|---|---|---|---|---|---|---|---|---|---|---|---|---|---|---|---|---|
| cm | g/kg | H2O | KCl | | dS/m | mg/kg | | | -----------------------cmolc/kg----------------------- | | | | | | | cmolc/kg of Clay | ------%------ | | |
| Profile 1—NEOSSOLO FLÚVICO Ta Eutrófico típico (Fluvisols) | | | | | | | | | | | | | | | | | | | |
| Ap (0–15) | 34.63 | 4.32 | 4.85 | 0.53 | 1.37 | 30.68 | 0.85 | 0.68 | 7.77 | 3.72 | 0.29 | 5.84 | 13.02 | 13.31 | 18.86 | 70.54 | 69 | 2 | 4 |
| 2C1 (15–34) | 17.91 | 3.98 | 3.78 | −0.19 | 0.23 | 55.34 | 0.21 | 0.34 | 4.15 | 1.25 | 0.71 | 4.88 | 5.95 | 6.66 | 10.83 | 55.77 | 55 | 11 | 3 |
| 2C2(34–55) | 16.72 | 4.30 | 3.96 | −0.34 | 0.20 | 83.80 | 0.13 | 0.41 | 5.02 | 2.04 | 0.49 | 2.63 | 7.59 | 8.08 | 10.23 | 59.68 | 74 | 6 | 4 |
| 3C3 (55–65) | 10.77 | 4.74 | 4.36 | −0.37 | 0.20 | 77.30 | 0.30 | 0.45 | 5.91 | 1.84 | 0.29 | 1.91 | 8.50 | 8.79 | 10.42 | 51.62 | 82 | 3 | 4 |
| 3C4 (65–85) | 7.95 | 5.03 | 4.61 | −0.43 | 0.23 | 62.06 | 0.25 | 0.41 | 6.18 | 1.98 | 0.21 | 1.31 | 8.83 | 9.04 | 10.14 | 44.44 | 87 | 2 | 4 |
| Profile 2—CAMBISSOLO HÁPLICO Ta Eutrófico típico (Cambisols) | | | | | | | | | | | | | | | | | | | |
| A (0–10) | 23.1 | 5.30 | 4.93 | −0.38 | 0.16 | 107.34 | 0.44 | 0.72 | 8.71 | 5.74 | 0.09 | 1.78 | 15.62 | 15.70 | 17.39 | 80.18 | 90 | 1 | 4 |
| AB (10–21) | 19.48 | 5.50 | 5.19 | −0.31 | 0.16 | 114.06 | 0.65 | 0.56 | 8.31 | 5.76 | 0.06 | 1.07 | 15.28 | 15.34 | 16.35 | 82.01 | 94 | 0 | 3 |
| BA (21–40) | 11.92 | 4.93 | 4.34 | −0.59 | 0.14 | 61.84 | 0.14 | 0.38 | 7.40 | 5.88 | 0.27 | 1.66 | 13.81 | 14.08 | 15.47 | 71.74 | 89 | 2 | 3 |
| B (40–60) | 8.36 | 4.91 | 4.16 | −0.75 | 0.13 | 48.39 | 0.13 | 0.31 | 6.26 | 6.49 | 0.32 | 1.52 | 13.19 | 13.51 | 14.71 | 64.50 | 90 | 2 | 2 |
| Profile 3—NEOSSOLO FLÚVICO Ta Eutrófico típico (Fluvisols) | | | | | | | | | | | | | | | | | | | |
| A1 (0–10) | 45.73 | 6.40 | 6.56 | 0.16 | 1.42 | 186.24 | 0.75 | 0.83 | 11.07 | 5.08 | 0.00 | 1.19 | 17.73 | 17.73 | 18.92 | 81.23 | 94 | 0 | 4 |
| A2 (10–20) | 21.53 | 4.87 | 4.57 | −0.30 | 0.42 | 149.03 | 0.07 | 0.29 | 6.56 | 2.14 | 0.18 | 2.04 | 9.06 | 9.24 | 11.10 | 71.92 | 82 | 2 | 3 |
| AC (20–50) | 5.83 | 5.22 | 4.76 | −0.46 | 0.74 | 100.84 | 0.04 | 0.36 | 8.15 | 2.63 | 0.14 | 1.47 | 11.18 | 11.32 | 12.65 | 66.01 | 88 | 1 | 3 |
| C (50–98) | 1.8 | 6.36 | 5.68 | −0.68 | 0.65 | 220.31 | 0.02 | 0.40 | 5.72 | 2.56 | 0.00 | 0.83 | 8.69 | 8.69 | 9.52 | 106.64 | 91 | 0 | 4 |
| Profile 4—LUVISSOLO CRÔMICO Órtico típico (Luvisols) | | | | | | | | | | | | | | | | | | | |
| A (0–10) | 27.04 | 5.50 | 5.43 | −0.07 | 0.23 | 112.05 | 0.95 | 0.74 | 6.85 | 2.75 | 0.47 | 2.71 | 11.29 | 11.75 | 13.99 | 110.02 | 81 | 4 | 5 |
| AB (10–20) | 16.66 | 5.10 | 4.90 | −0.20 | 0.15 | 96.36 | 0.47 | 0.68 | 6.65 | 2.49 | 0.14 | 2.57 | 10.29 | 10.44 | 12.87 | 84.78 | 80 | 1 | 5 |
| BA (20–33) | 10.83 | 4.82 | 4.32 | −0.50 | 0.13 | 23.06 | 0.06 | 0.28 | 7.64 | 1.78 | 0.24 | 2.56 | 9.76 | 10.00 | 12.32 | 42.60 | 79 | 2 | 2 |
| Bt (33–57) | 8.34 | 5.11 | 4.38 | −0.73 | 0.13 | 8.51 | 0.10 | 0.13 | 7.31 | 2.44 | 0.24 | 1.60 | 9.99 | 10.23 | 11.58 | 37.73 | 86 | 2 | 1 |
| BC (57–65) | 2.91 | 5.15 | 4.40 | −0.75 | 0.13 | 11.45 | 0.10 | 0.19 | 6.37 | 2.93 | 0.25 | 1.81 | 9.59 | 9.84 | 11.40 | 49.91 | 84 | 3 | 2 |
| Profile 5—PLANOSSOLO HÁPLICO Eutrófico típico (Planosols) | | | | | | | | | | | | | | | | | | | |
| A1 (0–10) | 35.82 | 6.70 | 7.06 | 0.35 | 0.42 | 199.54 | 0.88 | 0.11 | 8.11 | 2.16 | 0.00 | 0.53 | 11.27 | 11.27 | 11.80 | 102.09 | 95 | 0 | 1 |
| A2 (10–32) | 30.48 | 6.34 | 6.62 | 0.28 | 0.29 | 197.71 | 0.52 | 0.27 | 6.75 | 1.24 | 0.00 | 1.01 | 8.79 | 8.79 | 9.79 | 60.22 | 90 | 0 | 3 |
| A3 (32–44) | 14.97 | 5.16 | 4.87 | −0.29 | 0.22 | 21.01 | 0.10 | 0.34 | 3.07 | 2.40 | 0.08 | 2.12 | 5.91 | 5.99 | 8.03 | 44.04 | 74 | 1 | 4 |
| BA (44–64) | 10.42 | 4.46 | 4.25 | −0.21 | 0.23 | 27.05 | 0.09 | 0.25 | 2.15 | 1.51 | 0.29 | 2.27 | 4.00 | 4.29 | 6.26 | 45.03 | 64 | 7 | 4 |
| Bt (64–100) | 9.58 | 4.78 | 4.28 | −0.50 | 0.31 | 5.28 | 0.05 | 0.61 | 4.22 | 1.09 | 0.19 | 1.67 | 5.98 | 6.17 | 7.66 | 32.95 | 78 | 3 | 8 |
| Profile 6—NEOSSOLO LITÓLICO Chernossólico típico (Leptsols) | | | | | | | | | | | | | | | | | | | |
| A1 (0–28) | 43.74 | 6.35 | 6.43 | 0.09 | 0.32 | 233.19 | 0.72 | 0.41 | 9.81 | 0.83 | 0.10 | 1.43 | 11.77 | 11.87 | 13.20 | 102.51 | 89 | 1 | 3 |
| A2 (28–44) | 22.68 | 6.17 | 6.04 | −0.13 | 0.31 | 170.93 | 0.42 | 0.74 | 6.68 | 3.43 | 0.00 | 1.40 | 11.26 | 11.27 | 12.67 | 86.66 | 89 | 0 | 6 |
| CR (44–90) | 9.54 | 5.80 | 5.38 | −0.42 | 0.28 | 82.57 | 0.20 | 0.27 | 5.08 | 2.28 | 0.00 | 1.55 | 7.84 | 7.84 | 9.39 | 61.22 | 83 | 0 | 3 |
| Profile 7—ARGISSOLO VERMELHO-AMARELO Eutrófico típico (Acrisols) | | | | | | | | | | | | | | | | | | | |
| A (0–10) | 15.27 | 4.33 | 4.15 | −0.18 | 0.19 | 2.72 | 0.12 | 0.39 | 2.47 | 0.34 | 0.45 | 3.05 | 3.32 | 3.78 | 6.37 | 30.42 | 52 | 12 | 6 |
| BA (10–20) | 14.78 | 4.71 | 4.62 | −0.10 | 0.24 | 6.97 | 0.14 | 0.27 | 2.82 | 0.64 | 0.10 | 3.07 | 3.87 | 3.97 | 6.94 | 41.55 | 56 | 3 | 4 |
| Bt (20–50) | 4.48 | 4.38 | 4.43 | 0.05 | 0.13 | 0.72 | 0.06 | 0.24 | 2.78 | 1.44 | 0.28 | 2.62 | 4.52 | 4.80 | 7.15 | 11.75 | 63 | 6 | 3 |
| BC (50–75) | 4.03 | 4.30 | 4.37 | 0.07 | 0.13 | 1.34 | 0.05 | 0.26 | 2.22 | 1.77 | 0.40 | 2.42 | 4.31 | 4.70 | 6.73 | 11.91 | 64 | 8 | 4 |
| Profile 8—PLANOSSOLO HÁPLICO Eutrófico típico (Planosols) | | | | | | | | | | | | | | | | | | | |
| A (0–6) | 21.52 | 5.28 | 5.27 | −0.01 | 0.21 | 185.12 | 0.11 | 0.10 | 7.70 | 1.08 | 0.00 | 3.56 | 8.99 | 8.99 | 12.55 | 120.45 | 72 | 0 | 1 |
| BA (6–23) | 11.21 | 4.92 | 4.58 | −0.34 | 0.18 | 74.55 | 0.02 | 0.06 | 9.76 | 0.70 | 0.06 | 2.45 | 10.53 | 10.59 | 12.98 | 69.39 | 81 | 1 | 0 |
| Bt (23–43) | 10.77 | 5.22 | 4.29 | −0.93 | 0.16 | 62.64 | 0.02 | 0.08 | 11.66 | 1.88 | 0.18 | 1.82 | 13.64 | 13.82 | 14.98 | 78.82 | 88 | 1 | 1 |
| CR (43–60) | 7.18 | 5.24 | 4.35 | −0.89 | 0.14 | 65.16 | 0.02 | 0.10 | 7.76 | 1.65 | 0.10 | 1.29 | 9.52 | 9.62 | 10.81 | 75.24 | 88 | 1 | 1 |
| Profile 9—NEOSSOLO LITÓLICO Eutrófico típico (Leptsols) | | | | | | | | | | | | | | | | | | | |
| A (0–20) | 12.9 | 5.40 | 5.11 | −0.29 | 0.33 | 16.25 | 0.03 | 0.16 | 4.36 | 2.18 | 0.01 | 1.42 | 6.74 | 6.75 | 8.16 | 72.87 | 83 | 0 | 2 |

TOC—Total Organic Carbon; EC—Electrical Conductivity of the soil saturation extract; P—phosphorus; K+—potassium; Na+—sodium; Ca2+—calcium; Mg2+—magnesium; Al3+—aluminum; (H + Al)—potential acidity; SB—sum of bases; t—Effective cation exchange capacity; CEC—The cation exchange capacity; V—Base saturation; m—aluminum saturation; TSP—Exchangeable Sodium Percentage.

The delta pH values indicate that the soil shows a negative net charge. According to reference [30], the process of organic matter decomposition can acidify the environment through the release of humic substances. The use of fertilizers of animal origin, especially poultry manure, can reduce the pH since, according to reference [31], the mineralization of organic matter reduces the pH with the release of organic acids and $H^+$ ions in the process of organic material decomposition.

All soil profiles studied show low values of electrical conductivity of the saturation extract (EC), ranging from 0.13 dS/m to 1.42 dS/m, with the highest values being found in the surface horizons of Fluvisols (1.37 dS/m in the Hap of P1 and 1.42 dS/m in the A1 of P3) (Table 3). According to reference [32], salinization may occur in soils where anaerobiosis occurs, e.g., Gleysols, Planosols, and Fluvisols.

The low EC values indicate the non-occurrence of salinity problems in these soils since, according to reference [19], EC values above 4 dS/m and below 7 dS/m (at 25 °C) indicate soil salinity at some time of the year.

With regard to the total organic carbon (TOC), the highest contents were observed in the surface horizons, ranging from 12.9 g/kg in P9 to 45.73 g/kg in P3 (Table 3), decreasing with the depth, as commonly observed in tropical soils.

Overall, the physical and climatic conditions of the semi-arid region do not provide a higher production of organic waste sufficient to maintain the TOC. However, the study area consists of a mountain formation that differs from the predominant edaphoclimatic conditions of the semi-arid region. This scenario, associated with the soil management adopted (minimum preparation), contributes to the preservation of high TOC contents, which highlights the importance of adopting soil management practices that favor the increase in soil organic matter in the semi-arid region.

The phosphorus content (P) observed was higher in the surface environments, ranging from 16.25 mg/kg to 233.19 mg/kg (in horizon A of P9 and horizon A1 of P6, respectively) (Table 3). The high P contents can be associated with the conservationist practices adopted in the agroecosystems since, according to reference [33], management systems that increase the soil organic matter content also increase the P content in forms that are more readily available to crops by the action of organic acids resulting from organic matter decomposition, which blocks adsorption sites by coating Fe and Al oxides.

The calcium ($Ca^{2+}$) and magnesium ($Mg^{2+}$) contents ranged from 2.15 cmolc/kg to 11.66 cmolc/kg and from 0.34 cmolc/kg to 6.49 cmolc/kg, respectively, with the $Ca^{2+}$ values being always higher than the $Mg^{2+}$ values (Table 3). The minimum soil preparation system of the area favors the preservation of these nutrients. According to reference [34], studies related to agroforestry systems that employ minimum cultivation have reported significant gains of exchangeable bases in the soil solution due to the input of residues on the surface.

The aluminum contents ($Al^{3+}$) ranged from 0 to 0.71 cmolc/kg (Table 3), resulting in an aluminum saturation (m) that ranged from 0 to 12.97%, thus representing no aluminum toxicity to crops.

The cation exchange capacity (CEC) of the studied soils ranged from 6.07 cmolc/kg to 20.79 cmolc/kg (Table 3). These CEC values are maintained as a function of the sustainable management adopted in the area, which favors the increase and maintenance of organic matter. Only in profiles 1 and 3 is the CEC mainly occupied by $Na^+$ and $Ca^{2+}$ cations. Profile 7 showed CEC values well below the other profiles, which could be related to the amount of organic matter that contributes to increasing the CEC and the type and amount of clay minerals, since soils with the predominance of 2:1 clay minerals show a high CEC [35].

In all soil profiles, the base saturation (V) surpasses 50% (Table 3), making these soils eutrophic [19]. However, it should be noted that the $Na^+$ contents observed are greatly influenced by base saturation, contributing to an overestimation of fertility in these soils.

*3.5. Statistical Analysis*

The correlation matrix presents important information regarding the interaction of variables in the studied soils, with the sand fraction showing a high and negative correlation with the clay fraction, indicating an inverse increase trend between the two fractions (Table 4).

The silt fraction, in turn, showed a high and positive correlation with $K^+$. The pH ($H_2O$) showed a positive correlation with the elements P, $K^+$, $Ca^{2+}$, SB, and V, but a negative correlation with $Al^{3+}$ and (H + Al), highlighting that the pH increase favored nutrient release into the soil, reducing the availability of $Al^{3+}$ and (H + Al). The authors of reference [36] studied the physical and chemical attributes of a Ferralsol subjected to different agricultural uses in Martins—RN and found results that corroborate the present study.

In the factor analysis, the first two components explained 63.26% of the total data variance (Table 5).

Factor 1 explained 40.86% of the total variance and is related to nutrient availability, consisting of the attributes TOC, Ph ($H_2O$), P, $K^+$, $Ca^{2+}$, and CEC varying together and representing the variables that most influenced the distinction between soil classes. Adequate soil management greatly influences the input of TOC and other nutrients, whereas natural soil features favor the release of these nutrients.

**Table 4.** Correlation matrix between the physical and chemical variables of soil profiles in the Poção community, Martins/RN.

| | Sand | Silt | Clay | TOC | pH | EC | P | K$^+$ | Na$^+$ | Ca$^{2+}$ | Mg$^{2+}$ | Al$^{3+}$ | (H + Al) | SB | CEC | V |
|---|---|---|---|---|---|---|---|---|---|---|---|---|---|---|---|---|
| Sand | 1.00 | | | | | | | | | | | | | | | |
| Silt | **−0.48** | 1.00 | | | | | | | | | | | | | | |
| Clay | **−0.90** | 0.05 | 1.00 | | | | | | | | | | | | | |
| TOC | −0.05 | **0.73** | −0.30 | 1.00 | | | | | | | | | | | | |
| pH | **0.35** | 0.13 | **−0.46** | **0.48** | 1.00 | | | | | | | | | | | |
| EC | −0.21 | **0.65** | −0.08 | **0.52** | 0.25 | 1.00 | | | | | | | | | | |
| P | **0.35** | 0.22 | **−0.50** | **0.61** | **0.79** | 0.30 | 1.00 | | | | | | | | | |
| K$^+$ | −0.26 | **0.79** | −0.10 | **0.76** | **0.45** | **0.77** | **0.49** | 1.00 | | | | | | | | |
| Na$^+$ | −0.06 | **0.40** | −0.12 | 0.28 | 0.03 | **0.62** | 0.33 | **0.61** | 1.00 | | | | | | | |
| Ca$^{2+}$ | 0.10 | **0.34** | −0.29 | **0.45** | **0.51** | 0.31 | **0.54** | **0.46** | 0.22 | 1.00 | | | | | | |
| Mg$^{2+}$ | −0.08 | 0.14 | 0.01 | 0.14 | 0.19 | 0.22 | 0.11 | **0.37** | 0.32 | **0.34** | 1.00 | | | | | |
| Al$^{3+}$ | −0.33 | 0.06 | **0.34** | −0.20 | **−0.74** | −0.21 | **−0.49** | −0.20 | 0.15 | **−0.39** | −0.11 | 1.00 | | | | |
| (H + Al) | **−0.34** | **0.38** | 0.19 | 0.08 | **−0.68** | 0.17 | **−0.37** | −0.07 | 0.13 | −0.25 | −0.23 | **0.57** | 1.00 | | | |
| SB | 0.01 | **0.38** | −0.21 | **0.42** | **0.42** | **0.45** | **0.48** | **0.61** | **0.53** | **0.86** | **0.73** | −0.26 | −0.22 | 1.00 | | |
| CEC | −0.09 | **0.50** | −0.16 | **0.46** | 0.23 | **0.51** | **0.38** | **0.60** | **0.58** | **0.80** | **0.68** | −0.10 | 0.06 | **0.96** | 1.00 | |
| V | 0.31 | −0.02 | **−0.34** | 0.20 | **0.71** | 0.16 | **0.58** | 0.32 | 0.22 | **0.73** | **0.52** | **−0.53** | **−0.71** | **0.75** | **0.56** | 1.00 |

TOC: Total Organic Carbon; pH: hydrogenion potential; EC: electrical conductivity of the soil saturation extract; P: phosphorus; K$^+$: potassium; Na$^+$: sodium; Ca$^{2+}$: calcium; Mg$^{2+}$: magnesium; Al$^{3+}$: aluminum; (H + Al)—potential acidity; SB: sum of bases; CEC: cation exchange capacity; V—base saturation.

**Table 5.** Matrix of factor loadings after orthogonal rotation using the Varimax Method for the physical and chemical variables of soil profiles in the Poção community, Martins/RN.

| | Factor Loadings | |
|---|---|---|
| **Variables** | **Factor 1**<br>**Nutrient Availability** | **Factor 2**<br>**Inorganic Fractions** |
| Sand | −0.10 | 0.85 |
| Silt | −0.60 | −0.64 |
| Clay | 0.41 | −0.64 |
| TOC | **−0.77** | −0.19 |
| Ph | **−0.70** | 0.52 |
| EC | −0.67 | −0.45 |
| P | **−0.76** | 0.37 |
| K$^+$ | **−0.84** | −0.42 |
| Na$^+$ | −0.54 | −0.41 |
| Ca$^{2+}$ | **−0.76** | 0.14 |
| Mg$^{2+}$ | −0.47 | −0.13 |
| Al$^{3+}$ | 0.46 | −0.56 |
| CEC | **−0.78** | −0.25 |
| V | −0.68 | 0.44 |
| Eingevalue | 5.72 | 3.14 |
| Total Variance (%) | 40.86 | 22.40 |
| Total Variance Accumulated (%) | 40.86 | 63.26 |

For purposes of interpretation, significant factor loadings were considered ≥ 0.70.

Factor 2, with 22.40% of the total data variance, is composed of the sand fraction and is related to the inorganic soil fractions, being discriminated by the Fluvisol (P3), which showed a high total sand content of 800 g/kg in depth (Table 2), indicating that the relief of the study area allowed the formation of different soil classes with textural variation between them.

The projection diagrams of the vectors related to the chemical and physical attributes of the studied soils were generated using Principal Component Analysis (PCA). Factors 1 and 2 are responsible for the higher influence in the classification of variables that stood out in the distinction between the soil classes found (Figure 3A,B).

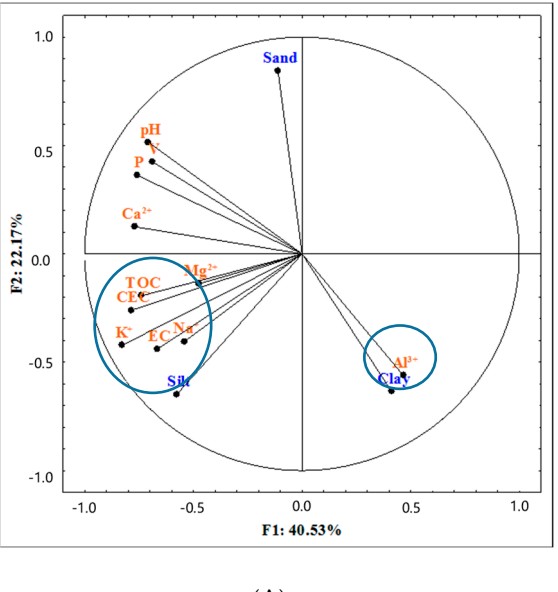
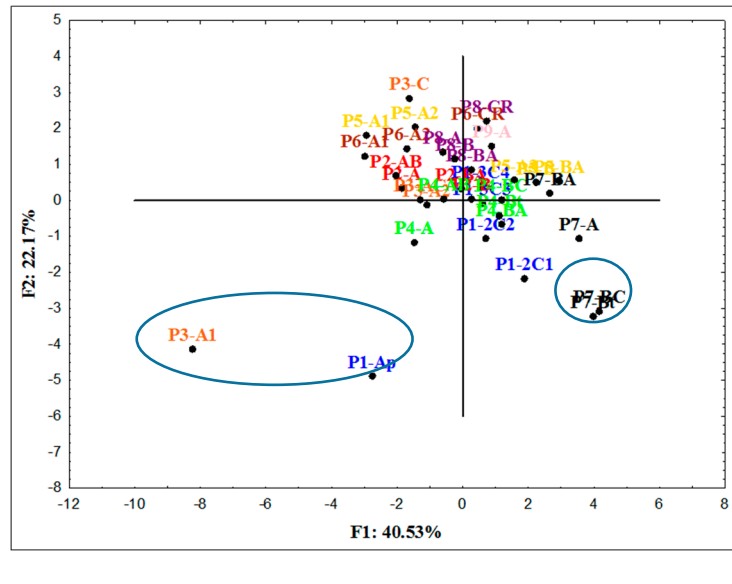

**(A)**                                                       **(B)**

**Figure 3.** Distribution of variables, in the circle of correlations (**A**) and distribution of the cloud of points representing the relationship between factors 1 and 2 (**B**). Note: P—profile.

It can be seen that sand, as well as the chemical variables of total organic carbon, Ph (H$_2$O), phosphorus, potassium, calcium, and the cation exchange capacity, were responsible for the differentiation of the classes observed as a function of the factor loadings and are located closer to the correlation circle.

When analyzing the diagrams, it is possible to recognize the formation of two distinct environments, with factor 1 explaining 40.86% of the cumulative variance, and factor 2 explaining 22.40%, amounting to a cumulative variance of 63.26% in the distribution of the variables chosen (Figure 3A,B).

One of the environments is formed using the Acrisol soil class (P7) (Figure 3B), with aluminum and the clay fraction being the factors that most contributed to the differentiation of this class (Figure 3A), indicating a naturally more acidic soil, with a more advanced pedogenesis in relation to the other classes found. The authors of reference [37] used multivariate analysis to differentiate environments and also determined that aluminum was one of the most discriminant variables for the Acrisol class.

The other environment is formed by the Fluvisol class (P1 and P3) in its surface environments (Figure 3B), with the chemical attributes of potassium, organic carbon, electrical conductivity of the soil saturation extract, sodium, cation exchange capacity, and the silt fraction constituting the factors that most stood out in the distinction of this soil class (Figure 3A). The authors of refence [38] studied the use of principal component analysis to cluster soil samples based on their particle size and chemical and mineralogical characteristics and observed that sediment deposition influenced the contents of the chemical attributes observed.

## 4. Conclusions

This study updates the soil classes observed in the study area, serving as a basis for better soil management according to the new classes found: Acrisols, Planosols, and Cambisols, with the relief being responsible for the difference between soil attributes. The influence of organic matter on the soil attributes demonstrates the importance of its maintenance. The clay fraction and aluminum are responsible for the distinction of the Acrisol class, whereas silt, potassium, sodium, total organic carbon, electrical conductivity of the saturation extract, and cation exchange capacity enable the differentiation of Fluvisols.

**Author Contributions:** Conceptualization, J.C.P. and P.K.P.F.; methodology, J.C.P. and P.K.P.F.; software, J.E.F.G. and J.D.d.C.; validation, P.K.P.F., J.C.P., R.O.B., J.D.d.C., J.E.F.G., E.F.d.S. and J.O.P.; formal analysis, P.K.P.F., J.C.P., R.O.B., J.D.d.C., J.E.F.G., E.F.d.S. and J.O.P.; investigation, J.C.P., P.K.P.F., J.E.F.G. and J.D.d.C.; resources, P.K.P.F., J.C.P., R.O.B., J.D.d.C., J.E.F.G., E.F.d.S. and J.O.P.; data curation, P.K.P.F., J.C.P., R.O.B., J.D.d.C., J.E.F.G., E.F.d.S. and J.O.P.; writing—original draft preparation, P.K.P.F., J.C.P., R.O.B., J.D.d.C., J.E.F.G., G.X.d.M., P.J.M., E.F.d.S., F.d.A.d.O., J.O.P., D.J.d.C.B., C.M.d.N., R.O.d.S., M.d.S.D., T.d.C.D.M. and A.G.R.A.; writing—review and editing, P.K.P.F., J.C.P., R.O.B., J.D.d.C., J.E.F.G., G.X.d.M., P.J.M., E.F.d.S., F.d.A.d.O., J.O.P., D.J.d.C.B., C.M.d.N., R.O.d.S., M.d.S.D., T.d.C.D.M. and A.G.R.A.; visualization, P.K.P.F., J.C.P., R.O.B., J.D.d.C and J.E.F.G.; supervision, J.C.P. and P.K.P.F.; project administration, J.C.P. and P.K.P.F.; funding acquisition, J.C.P. and P.K.P.F. All authors have read and agreed to the published version of the manuscript.

**Funding:** This research received no external funding.

**Data Availability Statement:** Not applicable.

**Acknowledgments:** The authors would like to thank the Coordenação de Aperfeiçoamento de Pessoal de Nível Superior (CAPES-BRASIL) for funding the doctoral scholarship and the Universidade Federal Rural do Semi-Árido who, through the Programa de Pós-Graduação em Manejo de Solo e Água, supported this research.

**Conflicts of Interest:** All the authors declare that they have no conflict of interest and have approved the manuscript and agreed to submit it to the Annals of Agricultural Sciences.

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
