# Peer review of "Characterization and Classification of Soils in Agroecosystems in a Moist Enclave in Northeastern Brazil"

_land, doi:10.3390/land12101881_

Round 1

Reviewer 1 Report

1. This study dug nine soil profiles to perform the morphological description and collect samples for the physical and chemical analyses in the community of Poção, Martins-RN. 

2. The description of land use on these digging sites is needed to explain the texture and structure of soil with their own geologic property.

3. Altitude, temperature and rainfall amount is suggested to be analyzed in the multivariate analysis for these weathering parameters could affect the soil structure. 

4. More analyses are suggested for the variation of each profile on particle size distribution. Especially, the discussion on percentage of gravel, sand and clay in different depth.

5. Except for multivariate analysis, more analyses are suggested on the variation of chemical attributes in different depth and on different profiles.

6. Deeper analysis and description of physical meaning is needed for table 5 and Figure 3.   

Author Response

"Por favor, verifique o anexo." na caixa se você carregar apenas um anexo.

Reviewer 2 Report

Materials and Methods include much information and may be better separated into subparts, such as Location of study area; Sample collection; Soil analyses; and Statistical analysis.

Author Response

"Please see the attachment." in the box if you only upload an attachment.

Reviewer 3 Report

Please, find my report attached. 

Author Response

Please see the attachment." in the box if you only upload an attachment. 

Round 2

Reviewer 1 Report

This paper has revised. I have no more comment.